Identification of key genes and long non-coding RNA associated ceRNA networks in hepatocellular carcinoma

Liu Jun 1 2
Li Wenli 3
Zhang Jian 1
http://orcid.org/0000-0001-6530-0539 Ma Zhanzhong 1
Wu Xiaoyan 4
http://orcid.org/0000-0002-9863-3732 Tang Lirui 2 5 tototang6@hotmail.com
1 Department of Clinical Laboratory, Yue Bei People’s Hospital , Shaoguan, Guangdong , China
2 Morning Star Academic Cooperation , Shanghai , China
3 Reproductive Medicine Center, Yue Bei People’s Hospital , Shaoguan, Guangdong , China
4 Community Healthcare Center , Shanghai, Shanghai , China
5 Shanghai JiaoTong University School of Medicine, Shanghai Ninth People’s Hospital , Shanghai , China
Uversky Vladimir
Electronic publication date: 2019 Nov 1
Publication date: 2019
Volume: 7
Electronic Location ID: e8021
Received 2019 Jul 8; Accepted 2019 Oct 10
Copyright: © 2019 Liu et al.
Copyright year: 2019
Copyright holder: Liu et al.
License: This is an open access article distributed under the terms of the Creative Commons Attribution License, which permits unrestricted use, distribution, reproduction and adaptation in any medium and for any purpose provided that it is properly attributed. For attribution, the original author(s), title, publication source (PeerJ) and either DOI or URL of the article must be cited.
License URL: https://creativecommons.org/licenses/by/4.0/

Keywords: Hepatocellular carcinoma, Competitive endogenous RNAs, Prognostic values, Molecular biological mechanisms, GEO database

Funding: The authors received no funding for this work.

==============================
Background

Hepatocellular carcinoma (HCC) is one of the leading causes of cancer-related deaths worldwide. Although multiple efforts have been made to understand the development of HCC, morbidity, and mortality rates remain high. In this study, we aimed to discover the mRNAs and long non-coding RNAs (lncRNAs) that contribute to the progression of HCC. We constructed a lncRNA-related competitive endogenous RNA (ceRNA) network to elucidate the molecular regulatory mechanism underlying HCC.

Methods

A microarray dataset (GSE54238) containing information about both mRNAs and lncRNAs was downloaded from the Gene Expression Omnibus database. Differentially expressed genes (DEGs) and lncRNAs (DElncRNAs) in tumor tissues and non-cancerous tissues were identified using the limma package of the R software. The miRNAs that are targeted by DElncRNAs were predicted using miRcode, while the target mRNAs of miRNAs were retrieved from miRDB, miRTarBas, and TargetScan. Functional annotation and pathway enrichment of DEGs were performed using the EnrichNet website. We constructed a protein–protein interaction (PPI) network of DEGs using STRING, and identified the hub genes using Cytoscape. Survival analysis of the hub genes and DElncRNAs was performed using the gene expression profiling interactive analysis database. The expression of molecules with prognostic values was validated on the UALCAN database. The hepatic expression of hub genes was examined using the Human Protein Atlas. The hub genes and DElncRNAs with prognostic values as well as the predictive miRNAs were selected to construct the ceRNA networks.

Results

We found that 10 hub genes (KPNA2, MCM7, CKS2, KIF23, HMGB2, ZWINT, E2F1, MCM4, H2AFX, and EZH2) and four lncRNAs (FAM182B, SNHG6, SNHG1, and SNHG3) with prognostic values were overexpressed in the hepatic tumor samples. We also constructed a network containing 10 lncRNA–miRNA–mRNA pathways, which might be responsible for regulating the biological mechanisms underlying HCC.

Conclusion

We found that the 10 significantly overexpressed hub genes and four lncRNAs were negatively correlated with the prognosis of HCC. Further, we suggest that lncRNA SNHG1 and the SNHG3-related ceRNAs can be potential research targets for exploring the molecular mechanisms of HCC.

Introduction

Hepatocellular carcinoma (HCC) is the third-largest cause of cancer-related deaths worldwide. There has been an upward trend in the incidence of HCC over recent decades. Importantly, the incidence and mortality of HCC have increased in certain non-traditional high-risk regions such as North America and several European regions. In the United States, HCC has already become the fastest-rising contributor for cancer-related deaths (Kulik & El-Serag, 2019). Several primary diseases, including chronic hepatitis C virus (HCV), hepatitis B virus (HBV), heavy alcohol consumption, and diabetes, have been confirmed as the main risk factors for HCC (Singal & El-Serag, 2015). Due to the high morbidity and mortality as well as the diversity of risk factors associated with HCC, there has been great emphasis on exploring early detection markers, novel therapeutic targets, and prognostic biomarkers. Nevertheless, factors such as high rates of recurrence and metastasis, complexity in treatment and the subsequent rise in financial burden has been a challenge for practitioners globally. Fortunately, the discovery of a variety of non-coding RNAs (ncRNAs), such as lncRNA and miRNA, has facilitated the medical researchers to comprehensively understand HCC.

Long non-coding RNAs (lncRNAs) belong to a class of RNAs that do not encode for proteins. These RNAs are more than 200 nucleotides in length, and are involved in regulating various cellular processes, such as chromatin remodeling, epigenetic modifications, and protein expression. Dysregulation of lncRNAs has been found to promote tumorigenesis and metastasis and is linked with poor prognosis in HCC (Abbastabar et al., 2018).

miRNAs are a type of ncRNAs that have been found to be evolutionarily conserved. They are endogenous and have an average length of 22-nt. miRNAs can fine-tune cellular processes in response to physiological and pathological changes by mediating post-transcriptional regulation of the target genes. The dysregulation of miRNAs and the subsequent anomalous interactions with other molecules have profound implications on the initiation, development, and therapy of HCC (Borel, Konstantinova & Jansen, 2012).

A growing body of evidence has shown that lncRNAs and miRNAs not only affect tumor progression separately but also cooperate with each other in modulating the cancer-related genes or pathways. Specifically, lncRNAs bind to mRNAs preventing their interaction with miRNAs and subsequently inhibit the miRNAs. On the other hand, miRNAs repress their target genes when aberrantly expressed. Therefore, the interaction between three types of RNAs results in the formation of a lncRNA–miRNA–mRNA network, which gives rise to competitive endogenous RNA (ceRNA). The crosstalk mediated by ceRNA assists in coordinating a large number of biological processes (BP). However, in pathological conditions, this can lead to the development of tumorigenesis (Karreth & Pandolfi, 2013).

In this study, we aimed at exploring the functional significance of lncRNA-related ceRNA networks in HCC based on the ceRNA network theory. We analyzed a microarray dataset containing information about both lncRNA and mRNA obtained from hepatic cancer tissues as well as non-tumor tissues to obtain the DElncRNAs and differentially expressed genes (DEGs). After the target miRNAs of the DElncRNAs as well as the potential upregulated mRNAs of miRNAs were predicted using online tools, the mRNAs that overlapped with the DEGs were selected for further analysis. The DEGs were used to construct a protein–protein interaction (PPI) network complex, and we used this information to identify 10 hub genes. Further, we performed survival analysis on all the hub genes and DElncRNAs. Eventually, 10 hub genes and four DElncRNAs with prognostic values were found to be upregulated in HCC. These hub genes and DElncRNAs as well as the predictive miRNAs were used to construct the ceRNA network.

Materials and Methods

Data acquisition and preprocessing

The GSE54238 expression profile was obtained from the Gene Expression Omnibus (GEO, https://www.ncbi.nlm.nih.gov/geo/), which was imputed on GPL16955 (Arraystar human lncRNA microarray V1-100309) and was found to contain 13 advanced HCC and 10 normal sample tissues. The downloaded raw data were preprocessed, including background adjustment, normalization, and gene biotype re-annotation.

Acquisition and analysis of expression profiles

The DElncRNAs and DEGs in the HCC and normal tissues were calculated using the limma package of R software (https://www.r-project.org/). mRNAs with a |log fold change (logFC)| ≥ 1 and an adjusted P-value < 0.05 were considered as the selection criteria of DEGs for subsequent analysis. The heat maps of lncRNAs and DEGs were drawn using the pheatmap package of the R software.

Construction of the ceRNA network

The interaction of miRNAs with DElncRNAs was predicted based on the file “Highly conserved microRNA families,” which was downloaded from a highly reliable online miRNA reference database, miRcode (http://www.mircode.org/) (Jeggari, Marks & Larsson, 2012). The prediction of target mRNAs of miRNAs was performed using three databases, miRDB (http://www.mirdb.org/) (Wong & Wang, 2015), miRTarBas (http://mirtarbase.mbc.nctu.edu.tw) (Hsu et al., 2014), and TargetScan (http://www.targetscan.org) (Park & Kim, 2013). We only selected the mRNAs that overlapped with DEGs for constructing the lncRNA–miRNA–mRNA network. The ceRNA network was visualized by Cytoscape (https://cytoscape.org/) (Shannon et al., 2003).

Functional annotation and pathway enrichment analysis of DEGs

Differentially expressed genes were analyzed by Gene ontology (GO) and kyoto encyclopedia of genes and genomes (KEGG) analysis on Enrich (http://www.enrichnet.org/) website to perform functional annotation and pathway enrichment.

Construction of the PPI network and screening for the hub genes

The protein–protein interaction network of the targeted DEGs was constructed using STRING (https://string-db.org/) and the Cytoscape. The non-interacting genes were excluded in order to simplify the PPI network. The top 10 genes with the highest degree of connection to the others were considered as hub genes based on the analysis using CytoHubba from Cytoscape.

Reconstruction of the ceRNA network

The hub genes and DElncRNAs were retrieved using the median value as the cut-off on the Gene Expression Profiling Interactive Analysis (GEPIA, http://gepia.cancer-pku.cn/) website, which is based on The Cancer Genome Atlas. Survival analyses were performed on these newly screened hub genes as well as the lncRNAs to obtain their prognostic values. Then, the hub genes and lncRNAs that showed statistical differences in the survival analysis were used to reconstruct a new ceRNA network. The expression of lncRNAs and hub genes with the most prominent statistical differences in survival was tested and verified using the UALCAN database (http://ualcan.path.uab.edu/). The hepatic expression of hub genes was examined using the Human Protein Atlas database (https://www.proteinatlas.org/). Correlation analyses were conducted to analyze the correlations between the expression of each lncRNA and its corresponding hub genes in the ceRNA network on GEPIA.

Results

Identification of DElncRNAs and DEGs

In total, 10 tissue samples from the control and 13 from the HCC tissues were available in the GSE54238 dataset. The expression profiles of the mRNAs and lncRNAs were calculated. According to the results analyzed by the limma package, 1,673 mRNAs (Fig. 1A) and 12 lncRNAs (Fig. 1B) were differentially expressed |(logFC)| ≥ 1 and adjusted P > 0.05) and they were defined as DEGs and DElncRNAs, respectively. Out of these, 768 mRNAs and 12 lncRNAs were over-expressed while 904 mRNAs and one lncRNA was downregulated.

Figure 1 Identification of DElncRNAs and DEGs.

(A) Volcano plots of the gene expression profile data. The blue dots represent down-regulated genes and the orange dots represent up-regulated genes. (B) Heat map of lncRNAs in HCC samples compared with normal samples.

Construction of the lncRNA-associated ceRNA network

Based on the results of the online prediction, we found that the miRNAs might interact with DElncRNAs (Table S1). The target mRNAs of the miRNAs are indicated with blue nodes (Fig. 2). Among all the predictive mRNAs, only the 126 mRNAs that also existed in the DEGs were selected to construct the first ceRNA network. Table S2 shows the details of the 126 mRNAs. The red nodes are DElncRNAs (GAS5, FAM41C, FAM182B, FAM138 B, FAM138C, FAM138D, FAM138E, UCA1, SNHG1, SNHG3, and SNHG6) and the green nodes represent the latently interactive miRNAs.

Figure 2 lncRNA-associated ceRNA network.

Red nodes represent the 11 DElncRNAs. Green nodes represent the 32 predictive miRNAs. Blue nodes represent the 126 predictive mRNAs.

Functional annotation and pathway enrichment analysis of DEGs

In order to thoroughly understand the properties and functions of the 126 DEGs in HCC development and progression, we applied functional annotation and pathway enrichment analysis on each of the DEGs from the ceRNA network. The GO analysis indicated that: (1) for BP, the DEGs from the ceRNA network were particularly enriched in positive regulation of protein localization on the membrane, positive regulation of protein insertion into mitochondrial membrane involved in apoptotic signaling pathway, positive regulation of mitochondrial outer membrane permeabilization involved in apoptotic signaling pathway, regulation of epithelial cell proliferation, regulation of nuclear-transcribed mRNA catabolic process, deadenylation-dependent decay, positive regulation of epithelial to mesenchymal transition, positive regulation of apoptotic processes, positive regulation of nuclear-transcribed mRNA catabolic process, and positive regulation of gene expression (Fig. 3A); (2) for cellular components (CC), the DEGs were significantly enriched in chromatin, nuclear chromosomes, nuclear chromatin, apical dendrite, cytoplasmic vesicle membrane, ESC/E(Z) complex, lipid droplet, chromosome, telomeric region, and centrosome (Fig. 3B); (3) for molecular functions, the DEGs were enriched in transcription regulatory region DNA binding, core promoter sequence-specific DNA binding, mRNA 3′-UTR AU-rich region binding, regulatory region DNA binding, RNA polymerase II activating transcription factor binding, core promoter binding, cadherin binding, L-amino acid transmembrane transporter activity, insulin-like growth factor receptor binding, and amino acid transmembrane transporter activity (Fig. 3C). KEGG analysis demonstrated that DEGs were particularly enriched in the cell cycle, microRNAs involved in cancer, central carbon metabolism in cancer, pentose phosphate pathway, PI3K-Akt signaling pathway, fluid shear stress and atherosclerosis, colorectal cancer, non-alcoholic fatty liver disease, small cell lung cancer, and cellular senescence (Fig. 3D).

Figure 3 GO and KEGG analysis of DEGs.

(A) Biological processes, (B) Cell components, (C) Molecular functions of GO analysis, (D) KEGG analysis.

Construction of the PPI network and identification of the hub genes

The PPI network complex contained 90 DEGs (Fig. 4A). We evaluating their degree of connection, and identified 10 hub genes (MCM4, CKS2, ZWINT, HMGB2, MCM7, KPNA2, E2F1, H2AFX, KIF23, and EZH2), which were all up-regulated in HCC. The hub genes indicated with red nodes (MCM4, ZWINT, MCM7, and KPNA2) had the strongest connections with others, while those indicated with orange (CKS2, HMGB2, KIF23, E2F1) and yellow nodes (EZH2, H2AFX) had moderate and weak connections, respectively (Fig. 4B).

Figure 4 PPI network of DEGs and hub genes.

(A) PPI network of DEGs. Blue nodes represent the interaction among DEGs. Only the 90 DEGs that interacting with other ones were demonstrated in the network. (B) A total of 10 Hub genes identified from the PPI network. From the red nodes to yellow ones, the connection degree of each molecule with others gradually decreases.

Survival analysis of hub genes and DElncRNAs

We performed survival analysis on the hub genes and DElncRNAs and found that 10 overexpressed hub genes (MCM4, MCM7, ZWINT, KPNA2, CKS2, KIF23, E2F1, HMGB2, EZH2, and H2AFX) were significantly related to poorer prognosis with worse survival times in HCC patients. Among these, KPNA2 and EZH2 had the highest prognostic value (Fig. 5). We used the information from the ICGC database to validate our analyses, and found them to be consistent with our results (Fig. S1). Four DElncRNAs (FAM182B, SNHG1, SNHG3, and SNHG6) were upregulated and were found to be negatively related to the prognosis of HCC, with SNHG3 having the best clinical predictive value (Fig. 6).

Figure 5 Overall survival analysis of the hub genes.

(A) MCM4. (B) MCM7. (C) ZWINT. (D) KPNA2. (E) CKS2. (F) KIF23. (G) E2F1. (H) HMGB2. (I) EZH2. (J) H2AFX. The overall survival of patients with highly expressed hub genes was significantly shorter than those with low expression.

Figure 6 Overall survival analysis of the DElncRNAs.

(A) FAM182B. (B) SNHG1. (C) SNHG3. (D) SNHG6. The overall survival of patients with highly expressed DElncRNAs was significantly shorter than those with low expression.

Validation of the expression of hub genes and DElncRNAs

Expression of the hub genes, as well as the DElncRNAs with prognostic significance, was tested using the UALCAN database in order to verify the differences in expression. Consistent with the results described before, all of the DElncRNAs (Fig. 7) and hub genes (Fig. 8) with prognostic significance were significantly overexpressed in HCC tissues compared with normal ones. Further details of the comparisons are shown in Table S3. In addition, comparative analysis of the hub genes from the ICGC database was consistent with that obtained from the UALCAN database (Fig. S2). Furthermore, hepatic expressions of the hub genes (MCM4, MCM7, ZWINT, KPNA2, CKS2, KIF23, E2F1, HMGB2, EZH2, and H2AFX) were visualized using immunohistochemistry based on the Human Protein Atlas. While MCM4, MCM7, and EZH2 were not detected in non-tumor tissues, they were moderately or highly expressed in tumor tissues. The genes, ZWINT, E2F1, HMGB2, KPNA2, and CKS2, were expressed at low or moderate levels in non-tumor tissues. However, the staining intensity or the range of positive distribution of all the other proteins increased in tumor samples, except for KPNA2 (Fig. 9).

Figure 7 Validation of lncRNA expression in UALCAN database.

(A) FAM182B. (B) SNHG1. (C) SNHG3. (D) SNHG6. Based on UALCAN, all DElncRNAs were significantly overexpressed in HCC tissues compared with the normal ones.

Figure 8 Validation of hub genes expression in UALCAN database.

(A) MCM4. (B) MCM7. (C) ZWINT. (D) KPNA2. (E) CKS2. (F) KIF23. (G) E2F1. (H) HMGB2. (I) EZH2. (J) H2AFX. Based on UALCAN, all hub genes were significantly overexpressed in HCC tissues compared with the normal ones.

Figure 9 Validation of hepatic expression of hub genes in the Human Protein Atlas database.

Expression of (A) MCM4, (C) KIF23, (E) MCM7, (G) E2F1, (I) ZWINT, (K) HMGB2, (M) KPNA2, (O) EZH2, (Q) CKS2, (S) H2AFX in non-tumor tissues. Expression of (B) MCM4, (D) KIF23, (F) MCM7, (H) E2F1, (J) ZWINT, (L) HMGB2, (N) KPNA2, (P) EZH2, (R) CKS2, (T) H2AFX in tumor tissues. Proteins encoded by MCM4, MCM7, ZWINT, CKS2, E2FI, HMGB2, and EZH2 were expressed higher in tumor than in non-tumor tissues. KPNA2 were expressed moderately in both groups. KIF23 and H2AFX were highly expressed in both tumor and non-tumor tissues.

Reconstruction of the lncRNA-associated ceRNA network

The newly screened hub genes and DElncRNAs were used to reconstruct a new ceRNA network. In total, 10 hub genes (KPNA2, MCM7, CKS2, KIF23, HMGB2, ZWINT, E2F1, MCM4, H2AFX, and EZH2), seven miRNAs (hsa-miR-107, hsa-miR-1297, hsa-miR-24-3p, hsa-miR-216b-5p, hsa-miR-217, hsa-miR-338-3p, and hsa-miR-107) and four lncRNAs (FAM182B, SNHG6, SNHG1, and SNHG3) were included (Fig. 10). The reconstructed network consisted of 10 lncRNA–miRNA–mRNA pathways, including FAM182B-hsa-miR-107-KIF23, FAM182B-hsa-miR-107-MCM7, SNHG6-hsa-miR-1297-KPNA2, SNHG6-hsa-miR-1297-CKS2, SNHG6-hsa-miR-24-3p-H2AFX, SNHG3-hsa-miR-338-3p-ZWINT, SNHG1-hsa-miR-217-EZH2, SNHG1-hsa-miR-23b-3p-HMGB2, SNHG1-hsa-miR-216b-5p-E2F1, and SNHG1-hsa-miR-216b-5p-MCM4 (Table 1). lncRNA SNHG1 had the highest number of connections with the hub genes.

Figure 10 The lncRNA–miRNA–mRNA ceRNA network constructed from hub genes and DElncRNAs.

Red nodes represent lncRNAs. Green nodes represent miRNAs. Blue nodes represent hub genes. A total of 10 lncRNA–miRNA–mRNA pathways were reconstructed here.

Table 1 Reconstruction of the lncRNA-associated ceRNA networks.

lncRNA	miRNA	mRNA	
SNHG1	has-miR-216b-5P	E2F1	
		MCM4	
	hsa-miR-217	EZH2	
	has-miR-23b-3P	HMGB2	
SNHG6	has-miR-1297	KPNA2	
		CKS2	
	has-miR-24-3P	H2AFX	
FAM182B	hsa-miR-107	MCM7	
		KIF23	
SNHG3	has-miR-338-3P	ZWINT	

Co-expression analysis of lncRNAs and hub genes from the ceRNA network

Correlation analyses showed that expressions of all the lncRNAs were correlated with that of their corresponding mRNAs (Figs. 11A–11F). SNHG1 had the strongest correlations with its hub genes as the correlation coefficient for E2F1, EZH2, HMGB2, and MCM4 being 0.67, 0.77, 0.72, and 0.7, respectively (Figs. 11C–11F). All of these indicate a strong correlation. SNHG3, the lncRNA with the most valuable clinical significance, also showed a strong correlation with ZWINT (R = 0.6, Fig. 11G). In terms of expression, FAM182B and SNHG6 were moderately related to their corresponding mRNAs with correlation coefficients ranging from 0.51 to 0.67.

Figure 11 Co-expression analysis of DElncRNAs from ceRNA network with related hub genes in HCC patients.

(A) lncRNA FAM182B with KIF23. (B) lncRNA FAM182B with MCM7. (C) lncRNA SNHG1 with E2F1. (D) lncRNA SNHG1 with EZH2. (E) lncRNA SNHG1 with HMGB2. (F) lncRNA SNHG1 with MCM4. (G) lncRNA SNHG3 with ZWINT. (H) lncRNA SNHG6 with CKS2. (I) lncRNA SNHG6 with H2AFX. (J) lncRNA SNHG6 with KPNA2. SNHG1 and SNHG3 were strongly correlatted with their hub genes. FAM182B and SNHG6 showed moderate correlation with corresponding hub genes.

Discussion

In this study, 10 hub genes and four lncRNAs were found to be overexpressed in HCC tissues. Meanwhile, survival analysis results confirmed that they were potential candidates for prognostic predictions in HCC patients. Among all the hub genes, KPNA2 and EZH2, which are tumor-promoting genes, have been extensively studied and were found to be the most effective predictors of survival time. SNHG3, a lncRNA that is relatively less known, had the highest predictive value for clinical HCC prognosis.

None of the hub genes were completely new in the field of HCC. It has been shown that HCC patients who had higher expression of KPNA2 in the nucleus have poorer prognosis and higher risk of recurrence (Jiang et al., 2014b). KPNA2 imparts proliferative and metastatic abilities to the HCC cells by aiding the entry of certain transcriptional factors, such as PLAG2, into the nucleus (Hu et al., 2014). MCM7 has been identified as a prognostic marker of HCC that promotes cell proliferation and tumorigenicity by suppressing cyclin D1 expression via the activation of the MAPK pathway. Through this mechanism, MCM7 initiates DNA replication, which is a key process in cell cycle progression (Qu et al., 2017). CKS2 is activated by the Wnt/β-catenin signaling pathway and promotes cell proliferation and inhibits apoptosis, which makes the HCC cells more aggressive as its overexpression is more frequently observed in poorly differentiated tumors (Li et al., 2018; Shen et al., 2010). KIF23 depletion inhibits tumor formation by inducing apoptosis in certain types of cancer, such as lung adenocarcinomas. However, there are diverse clinical research reports on HCC that contradict our results. According to a previous study, KIF23, which has two splice variants, V1 and V2, is overexpressed in HCC tissues, while it is almost undetectable in normal tissues. Patients with positive V1 expression, which is mainly located inside the nucleus, have better overall 5-year survival than those with negative results. However, KIF23 V2 had no correlation with overall survival (OS). Therefore, further in-depth studies should be undertaken to evaluate the function of KIF23 in HCC (Sun et al., 2015). HMGB2 overexpression was detected at both mRNA and protein levels in tumor tissues from HCC patients. It is associated with shorter OS, and is considered to be an independent prognostic factor. In an in vitro study, HMGB2 was found to promote HCC cell proliferation and impair drug sensitivity (Kwon et al., 2010). ZWINT protein is elevated in HCC tissues and shows correlation with tumor size and number. HCC patients with high expression of ZWINT tend to experience higher rates of tumor recurrence. ZWINT may enhance cell proliferation by disturbing the expression of cell cycle-related proteins, such as proliferating cell nuclear antigen (PCNA) and cyclin B1, which results in shorter OS (Ying et al., 2018). E2F1 protein is activated to a greater extent in tumor tissues compared to that in the normal liver or adjacent non-tumor tissues from the same HCC patients (Feng et al., 2015). In mice with copy number gains in E2F1, dosage-dependent spontaneous tumors only occurred in the liver, which indicates that E2F1 is specifically involved in the promotion of hepatic tumorigenesis (Kent et al., 2017). The survival analysis from another dataset, GSE14520, supported our results, which showed that MCM4 enrichment had significant association with the OS in HCC patients in the background of hepatitis B infection (Liao et al., 2018). H2AFX is upregulated and activated in HCC, as shown by the increase in the levels of total and phosphorylated H2AFX in tumor tissues (Evert et al., 2013). In addition, H2AFX together with another seven genes, constitutes a prognostic signature for HCC with C-indexes of 0.776, 0.745, and 0.789 for 1-, 3-, and 5-year OS, respectively (Zhang et al., 2019a). EZH2 is important in determining the character of hepatic tumors and immunohistochemistry results showed that this protein was positively stained in various types of malignant liver tumors. However, EZH2 has not been detected in benign hepatic diseases, such as adenomas or cirrhotic nodules (Hajósi-Kalcakosz et al., 2012). By inhibiting the natural killer cell-mediated eradication of tumor cells, EZH2 may create an immune microenvironment that is conducive to HCC cells (Bugide, Green & Wajapeyee, 2018).

We also constructed a lncRNA-related ceRNA network consisting of 10 possible lncRNA–miRNA–mRNA pathways with DEGs and DElncRNAs, in which the DElncRNAs might compete with the DEGs for binding to their target miRNAs. Through this process, the DElncRNAs can regulate the expression and function of DEGs without directly interacting with them. In all the pathways, lncRNA SNHG1 might potentially regulate most of the hub genes, including EZH2, HMGB2, E2F1, and MCM4. It is worth noting that our results demonstrated a close correlation between the expression of SHNG1 and the four mRNAs, which strengthens the possibility of mutual regulation between them. Since all the mRNAs and lncRNAs in this network originated from the same dataset, we believed the predictive regulatory relationships are convincing, and are worth investigating further.

SNHG3 was found to be overexpressed in HCC tissues as compared to normal hepatic samples in this study. In addition, lncRNA SNHG3 was the most valuable prognostic factor for HCC. Consistent with our finding, SNHG3 expression has been proven to be negatively associated with the OS, recurrence-free survival and disease-free survival in HCC patients (Zhang et al., 2016a). SNHG3 promotes metastasis and induces sorafenib resistance by inducing EMT via the miR-128/CD151 pathway in HCC cells (Zhang et al., 2019b). Although there have been no research investigations on the direct interactions between SNHG3 and its targeting miRNAs in HCC, several studies have found that SNHG3 forms a ceRNA network by sponging miRNAs in other cancers (Chen, Wu & Zhang, 2019; Huang et al., 2017; Wang et al., 2019; Zheng et al., 2019). For example, SNHG3 sponged miRNA-151a-3p (Zheng et al., 2019) and miR-196a-5p (Chen, Wu & Zhang, 2019) in order to promote cell growth, invasion, and migration in osteosarcoma.

miR-338-3p is downregulated in HCC and inhibits its progression in various ways including suppressing cell proliferation (Fu et al., 2012), countering Warburg effects (Nie et al., 2015) as well as inhibiting metastasis (Chen et al., 2017; Zhang et al., 2016b). A diverse array of molecules, such as mineralocorticoid receptors (Nie et al., 2015), HBV X (Fu et al., 2012), and circular RNAs (circRNA) (Li et al., 2019) have been validated as the direct upstream regulators of miR-338-3p in HCC. A novel mechanism indicating that miR-338-3p expression might be regulated by lncRNA SNHG3 has been identified. The downstream binding targets of miR-338-3p, including N-cadherin (Chen et al., 2017), MACC1 (Zhang et al., 2016b), HIF-1α (Xu et al., 2014), and CyclinD1 (Fu et al., 2012) have also been well studied in HCC. However, for the first time, this study has predicted ZWINT as a potential target of miR-338-3p.

ZWINT is upregulated in HCC samples as well as cell lines, and promotes cell proliferation by affecting the expression of cell-cycle proteins, such as PCNA, cyclin, and CDK1 (Ying et al., 2018). ZWINT has also been identified as a key biomarker in drug-induced liver injury, which indicates its possible involvement in drug metabolism (Cho et al., 2016). However, a study conducted in post-surgery HCC patients showed that ZWINT expression was decreased in tumor samples (Yang et al., 2018b). These contradictory results could be due to the different stages of the disease when these studies were carried out, and due to the diverse causes that lead to development of HCC. Several miRNAs, such as miR-127-3p and miR-1, have been found to directly or potentially target ZWINT and promote tumor progression (Jiang et al., 2014a; Xie et al., 2018). Here, we identified ZWINT as a target of miR-338-3p that might form a ceRNA network with lncRNA SNHG3 in HCC.

In this study, we found that lncRNA SNHG1 was negatively associated with HCC prognosis. This result is consistent with previous research, which showed that overexpression of SNHG1 correlated with larger tumor size, poor differentiation degree, and a worse clinical-stage (Zhang et al., 2016c). SNHG1 also had the most number of connections with the hub genes, which means that it can most likely regulate the expression of the four mRNAs found in this network through miRNAs. SNHG1 was found to exacerbate HCC by inhibiting miR-195 directly (Zhang et al., 2016d). However, to our knowledge, this study has reported for the first time that in HCC, hsa-miR-217, has-miR-216b-5p, and hsa-miR-23b-3p might be the other targets of SNHG1.

In HCC, decreased miR-217 expression is positively associated with vascular invasion, and TNM stage (Tian et al., 2017). miR-217 partially inhibits hepatic metastasis by binding to the 3′-UTR of MTDH mRNA (Su et al., 2014; Zhang et al., 2017), which encodes a protein that can induce cell growth and inhibit apoptosis (Li et al., 2015). However, miR-217 was also found to be overexpressed in HCC and is potentially involved in promoting hepatic disease (Jiang et al., 2017). miR-217 induces cancer stem cell-like phenotypes resulting in the activation of the Wnt pathway in HCC cells (Jiang et al., 2017). High miR-217 levels leads to fat deposition in hepatocytes (Yin et al., 2012). These conflicting results may be attributed to the difference in the cell lines used in the experiments and the different stages of the disease when the studies were conducted.

miR-216b-5p is an anti-tumor miRNA found in several cancers (He et al., 2019; Ren et al., 2019; Sun et al., 2019b; You et al., 2017). It is believed that miR-216b-5p inhibits HCV by suppressing host autophagy (Huang et al., 2018). Paradoxically, miR-216b-5p represses the expression of UDP-glucuronosyltransferase 2B, a group of enzymes that detoxifies carcinogens by binding to their 3′UTRs (Dluzen et al., 2016). In addition, few other studies have also explored how miR-216b-5p affects HCC progression.

Low hepatic miR-23b-3p expression is considered as a biomarker of HCC tumorigenesis and progression (He et al., 2018). Meanwhile, serum miR-23b-3p levels help in distinguishing between the HCV patients that are at high risk of progressing to HCC from those with low risk (Sun et al., 2019a). Mechanically, miR-23b-3p inhibits the epithelial-mesenchymal transition in HCC cells via sponging by lncRNA HOTAIR from ZEB1 (Yang et al., 2018a). Although it has been speculated that miR-23b-3p is also involved in fat metabolism, the conclusions are inconsistent. A study found that in HepG2 cells, miR-23b-3p inhibits the synthesis of ApoA protein, which is a risk factor for thrombotic diseases (Zeng et al., 2018), whereas another study reported that SIRT1, an enzyme that prevents lipid accumulation, was repressed by hsa-miR-23b-3p in hepatocytes (Borji et al., 2019).

Among all the mRNAs in the SNHG1-related ceRNA network, EZH2, a widely investigated oncogene, is highly expressed in HCC tissues and contributes to its progression (Sudo et al., 2005). Similar to the results in this study, EZH2 has been validated as a direct target of miR-217 in gastric cancer cells (Chen et al., 2015). Nevertheless, a study conducted in colorectal cancer cells has found that EZH2 promotes cell growth by directly binding to lncRNA SNHG1 without requiring mediation of miRNAs (Xu et al., 2018). This suggests that EZH2 expression might be regulated by SNHG1 through various pathways.

In HCC, the HMGB2 mRNA is overexpressed and serves as an independent prognostic factor for OS (Kwon et al., 2010). Similar to our predictive analysis, HMGB2 was found to promote gastric cancer cell autophagy and induce multidrug resistance by directly interacting with miR-23b-3p (An et al., 2015).

E2F1 is a transcription factor that has been known to promote HCC progression (Farra et al., 2017). Besides the findings in this study, miR-331-3p (Jin et al., 2019) and miR-34a (Han et al., 2019) have also been found to regulate E2F1 expression in HCC cell lines. However, precise interactions between E2F1 and miRNAs need further exploration.

MCM4 expression is inversely related to the OS in HBV-related HCC (Liao et al., 2018). The gene polymorphism of MCM4 also affects the chances of developing HCC as the patients who carry the MCM4 rs2305952 CC are less likely to have the disease (Nan et al., 2016). The regulatory mechanism of MCM4 in HCC has been barely explored as there is only one previous study reporting MCM4 as a latent target of miR-122-5p (Wen et al., 2018). The results of this study provide novel information that might aid further investigation.

In summary, we identified 10 mRNAs and four lncRNAs that may promote HCC progression. We also constructed the –-miRNA–mRNA ceRNA network in order to understand the latent regulatory mechanism between mRNAs and ncRNAs. The lncRNA SNHG1 and SNHG3-related ceRNAs can be selected as novel targets while exploring the molecular mechanism of HCC.

Conclusions

In this study, we identified 10 significantly overexpressed hub genes (MCM4, CKS2, ZWINT, HMGB2, MCM7, KPNA2, E2F1, H2AFX, KIF23, and EZH2) and four lncRNAs (FAM182B, SNHG1, SNHG3, and SNHG6) and found them to be negatively correlated with HCC prognosis. We also suggested that the lncRNA SNHG1 and SNHG3-related ceRNAs can be potential research targets for investigating the molecular mechanism of HCC.

Supplemental Information

Supplemental Information 1 Code used for difference analysis.

Click here for additional data file.

Supplemental Information 2 Code used for biotype.

Click here for additional data file.

Supplemental Information 3 Validation of hub genes expression in ICGC database.

(A) MCM4. (B) MCM7. (C) ZWINT. (D) KPNA2. (E) CKS2. (F) KIF23. (G) E2F1. (H) HMGB2. (I) EZH2. (J) H2AFX. All hub genes were significantly up-regulated in tumor tissues than in normal ones.

Click here for additional data file.

Supplemental Information 4 Overall survival analysis of the hub genes from ICGC database.

(A) MCM4. (B) MCM7. (C) ZWINT. (D) KPNA2. (E) CKS2. (F) KIF23. (G) E2F1. (H) HMGB2. (I) EZH2. (J) H2AFX. HCC patients with high expression of any of the hub genes had shorter overall survival time.

Click here for additional data file.

Supplemental Information 5 Supplemental Tables.

Click here for additional data file.

Additional Information and Declarations

Competing Interests

Author Contributions

Data Availability

The authors declare that they have no competing interests.

Jun Liu conceived and designed the experiments, performed the experiments, analyzed the data, prepared figures and/or tables, approved the final draft.

Wenli Li conceived and designed the experiments, performed the experiments, prepared figures and/or tables, approved the final draft.

Jian Zhang analyzed the data, contributed reagents/materials/analysis tools, prepared figures and/or tables, authored or reviewed drafts of the paper, approved the final draft.

Zhanzhong Ma analyzed the data, contributed reagents/materials/analysis tools, authored or reviewed drafts of the paper, approved the final draft.

Xiaoyan Wu analyzed the data, contributed reagents/materials/analysis tools, authored or reviewed drafts of the paper, approved the final draft.

Lirui Tang conceived and designed the experiments, authored or reviewed drafts of the paper, approved the final draft.

The following information was supplied regarding data availability:

The lncRNAs and mRNAs expression profile are available at the GEO database: GSE54238.

The code is available in the Supplemental Files.

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
