# Peer review of "Identification of key genes and long non-coding RNA associated ceRNA networks in hepatocellular carcinoma"

_PeerJ, doi:10.7717/peerj.8021_

## Round 0.1 · original submission · Major Revisions

Please address all the critiques of both reviewers and revise your manuscript accordingly.

Reviewer 1 ·

Basic reporting

1-Figure legends could be more improved and have more information.
2- Legend of figure 10 could be more summarized and well provided (same information can be mentioned one time, for example it can be explaineed that these information are similar for these genes).

Experimental design

no comment

Validity of the findings

In conclusion the limitations of study may be explained, for example, small samples size.

Additional comments

1- In line 124, what is ad before DEGs?

2-In result section: " DElncRNAs and DEGs identification" should be improved and explain more details.

3-In result section: in "Construction of the lncRNA-associated ceRNA network", the 126 mRNAs can be listed in a supplementary files with informations.

4- In lines 176, 185, and 188 for BP, CC, MF abbreviations, their full names should be described for the first time.

5- In line 210, it could mentioned, for example, poor prognosis with worse survival time.

6- In the section of "Validation of the expression of hub genes and DElncRNAs", it was mentioned that hepatic expressions of the hub genes were visualized by immunohistochemistry on The Human Protein Atlas..... While information about 10 genes were not mentioned in text, they can be seen in figure 10.

7-In section of "Validation of the expression of hub genes and DElncRNAs", it is suggested to summarize information about TPM of normal vs tumor samples (from this webserver) for the validated genes in a table or in figures 8 and 9 to see the significant value of differential mRNA expression between these two groups.
8- Regarding Figure 8 and 9, it was better to mention the p value given by UALCAN web server (for most genes p value are approximately < 1E-12)

9-In the section of "Reconstruction of the lncRNA-associated ceRNA network", it was better to summarize all information in a table.

10- In section of "Co-expression analysis of lncRNAs and hub genes from ceRNA network", it is suggested to mention the correlation, for example, as weak, middle, strong, and very strong.

11-Line 262, it can be explained while the differential expression of SNHG3 was not so high ( If authors consider TPM), it showed poorer prognosis and the worst survival time. Therefore, authours may conclude other information for this.

12- In line 295, the hsa-miR-23b-3 should be miR-216b-5p as seen in figure 11?

Reviewer 2 ·

Basic reporting

no comment

Experimental design

Simple Bioinformatic analysis.

Validity of the findings

More literature validation is needed.

Additional comments

In this paper, DEGs and DElncRNA were extracted using a set of expression profile data; miRcode was used to predict miRNAs; after functional enrichment of genes, several Hub genes were extracted using protein interaction (PPI) network data; GEPIA was used for survival analysis; finally, UALCAN was used for prognosis worthy of verification.

Major issues:
1.Too many predictors may lead to skewed results, therefore, more references should be provided for validation.
2.One set of data is not enough to justify the results.
3.The competing endogenous between DEGs and DElncRNAs should be briefly described.
4.Hub genes should be associated with HCC in other literature, the relationship should be added in the Results part.
5.A protein interaction network (STRING) is incomplete.
6.The English issues must be revised carefully.

Minor issues:
1.lncRNA-miRNA-mRNA network have little to do with this article.
2.The description at lines 85-87 is confusing.
3.At line 99-101, the phrase “not only ...but also..” lacks “not only”.
4.The line 42 in Abstract is wrong in English.
5.The line 84 is wrong in English.

---

## Round 0.2 · Minor Revisions

Please address remaining issues pointed by the reviewer

Reviewer 1 ·

Basic reporting

English language should be improved. There are some mistakes.

Experimental design

no comment

Validity of the findings

no comment

Additional comments

1- Lines 171-174: the analytical results conducted by the limma package, a total of 1673 mRNAs (Fig. 1A) and 11 lncRNAs (Fig. 1B) were differentially expressed (|logFC)| ≥1.0 and adjusted P less than 0.05) and they were defined as DEGs and DElncRNAs respectfully. Of these, 768 mRNAs and 12 lncRNAs were over-expressed.
How can we see 12 lncRNAs out of 11 lncRNAs? These are needed to be corrected.

2-Lines 215-218:Hub genes with red nodes (MCM4, ZWINT, MCM7, and KPNA2) had the strongest connections with others, while those with orange (CKS2, HMGB2) and yellow nodes (EZH2, H2AFX) had moderate and weak connections, respectively (Fig. 4B).
Other genes with orange color were not included in this part.
3-Line 260: (R=0.7) was mentioned in this line and next line.
4- Line 329 and 331: Consistent with our finding was repeated.
5- Line 348: predicted word was repeated.

---

## Round 0.3 · accepted · Accept

Thank you for addressing remaining critiques. I am glad to inform you that since all the required changes were made, your manuscript is acceptable now.